# Enhancing Bioactive Components of *Euryale ferox* with *Lactobacillus curvatus* to Reduce H_2_O_2_-Induced Oxidative Stress in Human Skin Fibroblasts

**DOI:** 10.3390/antiox11101881

**Published:** 2022-09-23

**Authors:** Yanbing Jiang, Shiquan You, Yongtao Zhang, Jingsha Zhao, Dongdong Wang, Dan Zhao, Meng Li, Changtao Wang

**Affiliations:** 1Beijing Key Laboratory of Plant Resource Research and Development, College of Chemistry and Materials Engineering, Beijing Technology and Business University, Beijing 100040, China; 2Institute of Cosmetic Regulatory Science, Beijing Technology and Business University, Beijing 100040, China

**Keywords:** *Euryale ferox*, *Lactobacillus curvatus* fermentation, bioactive components, oxidative stress and antiaging

## Abstract

This study investigated the effects of *Lactobacillus curvatus* fermentation on the oxidative stress attenuating effects of *Euryale ferox* on H_2_O_2_-induced human skin fibroblasts (HSF). The results showed that *Lactobacillus curvatus* fermentation (i) increases the content of the various bioactive components of *Euryale ferox* and is found to have smaller molecular weights of polysaccharides and polypeptides; (ii) increases the overall intracellular and extracellular antioxidant capacity of H_2_O_2_-induced HSF while reducing reactive oxygen species (ROS) levels. Superoxide dismutase (SOD), glutathione peroxidase (GSH-Px), and catalase (CAT) all showed simultaneous increases in activity. Aside from that, the Nrf2 and MAPK signaling pathways are activated to regulate downstream-associated proteins such as the Bax/Bcl-2 protein ratio, matrix metalloproteinase 1 (MMP-1) activity, and human type I collagen (COL-1). These results suggested that the fermentation of *Euryale ferox* with *Lactobacillus curvatus* enhances its antioxidant capacity and attenuates apoptosis and senescence caused by oxidative stress.

## 1. Introduction

Oxidative stress is a condition in which the intracellular pro-oxidant–antioxidant balance is upset, resulting in excess reactive oxygen species (ROS) that damages DNA and proteins either directly or indirectly. This condition impairs cell viability and is a significant factor in aging and disease [1]. Antioxidants are substances that stop, slow down or minimize oxidative damage, and studies have shown that antioxidants can mitigate the harmful effects of oxidative stress. The hunt for plant-based antioxidants as opposed to synthetic antioxidants has recently gained popularity. Natural antioxidants can be utilized in dietary supplements, functional foods, cosmetics, medicine, and food preservation [2]. The body is shielded from free radicals by antioxidants, which also reduce the progression of many diseases. The body requires exogenous regulation and supplementation to minimize oxidative stress because the endogenous antioxidant system has a finite capacity for antioxidants. Consequently, there is a requirement to create and utilize efficient antioxidants [3]. At present, o-quinones, flavonoids with phenolic hydroxyl structures and other phenolic chemicals make up the majority of the natural antioxidants. Therefore, the natural phenolic compounds, flavonoids, and other biological activities in *Euryale ferox* have antioxidant and other physical activities, making them natural oxidants with good prospects for cosmetic applications.

The only species of the genus Euryale endemic to Eastern Asia is an aquatic plant called *Euryale ferox* which can be found in China, India, Korea, Japan, Southeast Asia, and other countries in the region. In China, *Euryale ferox* are mainly grown in Jiangsu, Shandong, Hunan, Hubei, and Anhui provinces. *Euryale ferox*, a medicinal and edible plant, contains a variety of polyphenols, flavonoids, amino acids, and other compounds that have pharmacological effects such as antioxidants, delaying aging, and anti-fatigue [4,5,6].

To increase the biological activity and release of phenolic components in grains, cereals, pulses, fruits, teas, and agricultural goods, microbial fermentation is now a widely utilized biotechnological processing technique [7]. It is clear that the type of fermentation impacts the potential bioactive content of cereals because it causes multiple biochemical changes that increase the levels of the numerous bioactive compounds in cereals [8]. The superior fermentation properties of lactic acid bacteria enhance the flavor of the fermented substrate while effectively maintaining and even improving its functional qualities, which is why they are commonly utilized in food production [9]. In cosmetics, lactobacilli are also superior in their antioxidant capacity, reducing UV damage, enhancing the skin’s resistance to photoaging, and repairing the skin barrier [10]. After blueberry juice was fermented by Lactobacillus plantarum, it was discovered that its total phenol levels and antioxidant activity were significantly increased [11]. Ravish Bhat and colleagues used Lactobacillus plantarum (NCIM 2912) to ferment guava, which increased the fruit’s overall phenolic content to 66.48%, significantly boosting its antioxidant capacity [12]. Because of the increased number of phenolic compounds and flavonoids produced during fermentation as a result of microbial hydrolysis processes, fermentation has the ability to improve antioxidant activity [13]. *Lactobacillus curvatus* is a lactic acid bacterium with good antioxidant properties which is mostly linked to chicken and fermented meat products, with no reports of plant fermentation [14]. However, it has been discovered to possess a number of genes linked to the generation of bacteriocin and the consumption of carbohydrates, which may give it powerful antibacterial and carbohydrate fermentative properties [15].

Research on *Euryale ferox* at home and abroad is not yet detailed and is limited to studies on nutritional composition, quality of origin, chemical composition, pharmacological effects, and clinical applications. As far as we know, no fermentation nor any other biotechnological procedure has been used to study the biological activity of *Euryale ferox*. The aim of this study was to assess changes in the biological activity of *Euryale ferox* before and after fermentation by *L. curvatus* and investigate its antioxidative stress damage effects at the cytotoxic, biochemical, cellular, and molecular levels.

## 2. Materials and Methods

### 2.1. Materials

*Euryale ferox* purchased online from JianZhiJiang Pharmacy; *Lactobacillus curvatus* extracted from the traditional snack bean sour liquid of Luoyang, Henan Province; human skin fibroblasts (HSF) were purchased from Peking Union Medical College Hospital; *Lactobacillus curvatus* was obtained from the traditional Chinese food mung bean sour liquid.

PBS, DMEM medium, newborn calf serum, streptomycin (100 mg/L), penicillin (1 × 10^5^ U/L), and 0.25% trypsin (including EDTA) were purchased from GIBCO Life Technologies, Shanghai, China; ascorbic acid and 1,1-diphenyl-2-trinitrophenylhydrazine (DPPH) were purchased from Alfa Aesar, China; Chemical Co., Ltd., Shanghai, China; Western and IP cell lysate, PMSF and ROS Detection Kits were purchased from Biyuntian Biotechnology Co., Ltd., Shanghai, China; CCK-8 Cell Proliferation and Toxicity Detection Kit (Biorigin (Beijing) Inc., Beijing, China; Total Glutathione Peroxidase (GSH-Px) Assay Kit with Nicotinamide Adenine Dinucleotide Phosphate (NADPH), Catalase Assay Kit, ROS Detection Kit and Cell Counting Kit, Total Antioxidant Capacity Test Kit (ABTS method) were purchased from Beyotime Biotechnology Co., Ltd., Shanghai, China; Human Elastase ELISA Kit, Human Collagen Type I ELISA Kit, Human Total Matrix Metalloproteinase 1 (MMP-1) ELISA Kit and Human NRF2 ELISA Kit were purchased from Cusabio (https://www.cusabio.cn/, accessed on 23 March 2022), Wuhan, China.

### 2.2. Preparation of Microorganism Inoculum

*L. curvatus* was reactivated in MRS broth for 12 h at 37 °C. Culturing was then conducted under the same conditions until the Bacterial suspensions’ absorbance at 600 nm reached 1.000. The culture served as the inoculum of the fermentation process [16].

### 2.3. Fermentation with Lactobacillus Curvatus

*Euryale ferox* was physically dried at 60 °C before being crushed. It was then filtered through a 50-mesh sieve, added to deionized water at a ratio of 1:15 (mg/mL), and sterilized at 120 °C for 15 min. *L. curvatus* suspension was added at an inoculum of 1:15. The fermentation process was carried out at 37 °C for 24 h. The control used was an aqueous extract of unfermented *Euryale ferox*. After fermentation, the fermentation extract is centrifuged, the precipitate is discarded, and the resulting supernatant is freeze-dried.

### 2.4. Euryale ferox Bioactive Component Assay

All phenolic contents were determined using the Folin–Ciocalteu reagent according to the method described [17]. Flavonoids were determined according to the method used in [18] as a reference. The protein content of *Euryale ferox* was determined using Coomassie blue staining.

### 2.5. Determination of Molecular Mass of Polypeptides

They were assayed using the method in [19] as a reference.

### 2.6. Determination of Molecular Mass of Polysaccharides

The solution to be analyzed was first diluted to 3 mg/mL and filtered through a 0.20–0.22 mm aqueous membrane, then packed into sample vials and kept in a freezer at 4 °C for the high-performance gel permeation chromatography (GPC) procedure to determine the molecular weight. The columns used included Ultrahydrogel Guard Column 125 (6 mm × 40 mm), Ultrahydrogel 250 Column (7.0 mm × 300 mm), and Ultrahydrogel 120 Column (7.0 mm × 300 mm). The samples were eluted at a specific rate (0.8 mL/min; Det: 50 °C; Col: 60 °C) and temperature (0.2 M sodium nitrate with sodium diazide, 0.02%).

### 2.7. FT-IR Analysis of Polysaccharides

They were assayed using the method in [19] as a reference.

### 2.8. DPPH and ABTS Scavenging Activity

They were assayed using the method in [19] as a reference.

### 2.9. Establishment of Cell Culture and H_2_O_2_ Damage Model

#### 2.9.1. Cell Culture

HSF were treated with DMEM complete media with 10% fetal bovine serum and 1% penicillin–streptomycin. The cells were then cultivated for two days at 37 °C with a CO_2_ level of 5% in an incubator at a constant temperature and humidity.

#### 2.9.2. H_2_O_2_ Damage Model

In this experiment, the cytotoxicity and protection against H_2_O_2_ damage of *Euryale ferox* water extract and *Euryale ferox* fermentation broth were detected using the CCK-8 method. HSF (8 × 10^4^–1 × 10^5^ cell/mL) were first cultured in a 96-well plate, covering all wells for 8–12 h. The sample was divided into three groups: blank, H_2_O_2_ group, and sample group. An amount of 100 µL of basal medium containing samples of different concentrations was added to each of the sample wells and 100 µL of basal medium was added to each of the control wells. After the cells had been incubated for 24 h, each well received a 12 h stimulation with H_2_O_2_ solution, and the absorbance was measured at 450 nm.

### 2.10. Cell Viability Determination

Each well in the 96-well plate received 100 μL of HSF (8 × 10^4^−1 × 10^5^ cells/mL) and was pre-incubated for 24 h in a cell incubator. Next, 100 μL of *Euryale ferox* samples was introduced to certain wells in place of the culture medium, while 100 μL of distilled water was added to the remaining wells. The absorbance was measured at 450 nm after each well received 100 μL of CCK-8 (CCK-8 Cell Proliferation and Toxicity Detection Kit, Liji Biotechnology Co., Ltd., shanghai, China) for 2 h. The following formula was used to determine cell viability: Cell viability (%) = (*A* − C)/(*B* − C) × 100%. *A*: Absorbance of sample wells; *B*: Absorbance of blank (no sample added); C: Absorbance of blank control (no cells added).

### 2.11. Determination of Total Antioxidant Capacity and ROS

Testing with Total Antioxidant Capacity Test Kit (ABTS method).

Each well in the 6-well plate received 2 mL of HSF and 2 mL of DCFH-DA (DCFH-DA: DMEM = 1:1000) over 24 h. Four groups were created: blank, H_2_O_2_, positive control, and sample. For the sample wells, 2 mL of *Euryale ferox* sample was placed in each. For the positive control wells, 2 mL of VC sample was placed in each. H_2_O_2_ was then used to irradiate the H_2_O_2_ group, positive control, and sample groups. To stimulate the cells, the sample group also received VC and *Euryale ferox* samples at varying doses. After 24 h of incubation, the cells were collected, and the fluorescence intensity of ROS in HSF cells treated with *Euryale ferox* samples was identified at 488 nm and 525 nm, respectively.

### 2.12. Antioxidant Enzymes and ELISA

Superoxide dismutase (SOD), catalase (CAT), and glutathione peroxidase (GSH-Px) were assayed using the method in [19] as a reference.

MMP-1, human type I collagen (COL-I), Nrf-2, Keap1, heme oxygenase 1 (HO-1), NAD(P)h: quinone oxidoreductase 1 (NQO-1), GCLC, BAX, and BCL-2 were measured according to the ELISA kit instructions.

### 2.13. Statistical Analysis

Each sample underwent three technical replicates and analyses across all experiments, which were each completed three times. Mean and standard deviation were used for analysis. The data were analyzed to identify pairs that differed significantly using one component analysis of variance (ANOVA) and Dunnett tests. GraphPad Prism 9 (GraphPad Software, Inc., La Jolla, CA, USA) was used to conduct the statistical analysis.

## 3. Results

### 3.1. Changes in Content of Bioactive Components in Euryale ferox

As shown in Table 1,fermentation with *L. curvatus* enhanced the release of bioactive compounds in *Euryale ferox*. The results showed that the fermentation of *Euryale ferox* with *L. curvatus* could effectively (*p* < 0.05) enhance its total phenolic content (TPC). The TPC concentration obtained was 0.052 (mg/mL) in LCF, which was 1.268-fold higher than that of UF. The total flavonoid contents in *Euryale ferox* after fermentation showed a similar tendency. The total flavonoid content increased significantly, from 1.358 to 1.402 (mg/mL). The total protein, polypeptide, and polysaccharide contents in unfermented *Euryale ferox* were 4.32, 0.399, and 1.657 mg/mL, respectively. After fermentation by *L. curvatus*, we observed an increase in polysaccharide and total protein content in *Euryale ferox* compared to UF.

### 3.2. Spectroscopy Analysis of Euryale ferox

#### 3.2.1. Polypeptide and Polysaccharide Molecular Weights

The effectiveness of *Euryale ferox* is affected by the molecular weight distribution and structure of its active constituents, including polypeptides and polysaccharides. From the high-performance liquid chromatography (HPLC) characterization of the two samples, we found that the molecular weight of polypeptides in LCF was 2539.43 Da, up to 92.79%. After fermentation, the molecular weight of the polypeptides was partially reduced, 36.23% less than 2000 Da (Figure 1 and Table 2). The GPC detection of the molecular weights of polysaccharides in UF and LCF is shown in Figure 2 and Table 3. It is evident that there is a small number of polysaccharides at 10^3^ Da in *Euryale ferox*, but most are concentrated in the range of 10^6^ Da. The results showed that the molecular weights of both the polypeptides and polysaccharides of *Euryale ferox* were reduced after fermentation.

#### 3.2.2. Characterization of Molecular Structures

As a common component of traditional medicines, polysaccharides serve a number of purposes, including anti-inflammatory, antioxidant, and lipid peroxidation inhibition functions [20]. The fundamental structure of polysaccharides can be studied using infrared spectroscopy, in which the peaks correspond to the vibrations of the various functional groups in the molecule. The findings of the analysis of the infrared spectra of *Euryale ferox* polysaccharides before and after fermentation are displayed in Figure 3. The IR spectra of *Euryale ferox* polysaccharides before and after fermentation were similar. All polysaccharides had absorption peaks of about 3400 cm^−1^ and 2930 cm^−1^, which correspond to the O-H and C-H stretching vibrations, respectively. Both polysaccharides contain acetamido, as evidenced by the C=O stretching vibration of acetamido-CONH_2_, the specific absorption peak of acetamido, which exhibits an absorption peak at 1654 cm^−1^. The absorption peak at 1654 cm^−1^ comes from the C=O stretching vibration of acetamido-CONH_2_, which is a specific absorption peak for acetamides, indicating that both polysaccharides contain acetamides. The relatively large absorption peak between 1250 cm^−1^ and 950 cm^−1^ is a characteristic absorption peak where the ether bond (C-O-C) and hydroxyl groups of the pyranose ring are important. Both polysaccharides, UF and LCF, exhibit high absorption in this area, proving that they both contain pyranose ring structures both before and after fermentation [21,22,23,24]. However, there was a clear difference in the intensity of the peaks of the two samples, with the absorption peak of LCF being more intense, particularly at 1087 cm^−1^ and 1021 cm^−1^, suggesting a difference in the content of the functional groups of the two polysaccharides. This might be connected to factors including the types of enzymes produced by *L. curvatus* throughout the fermentation and enzyme activity and persistence during fermentation.

### 3.3. Scavenging Free Radicals

The results of the DPPH radical scavenging activity of all samples at varying concentrations are shown in Figure 4a. UF and LCF demonstrated DPPH radical scavenging ability in a concentration-dependent manner, as shown in Figure 4a. The two samples’ clearance increased between a concentration of 600 mg/mL and 800 mg/mL. Both samples showed similar scavenging effects on ABTS as DPPH (Figure 4b).

### 3.4. Effects of UF and LCF on HSF Cell Viability

The effects of various drugs on cell viability can be determined by cytotoxicity assay. Figure 5a shows the toxicity of UF and LPF to cells. The results showed that the two systems were less toxic at 5.0 mg/mL, 10.0 mg/mL, and 20 mg/mL, and cell viability was higher than 90% among the three doses. Additionally, Figure 5 illustrates how both systems affect the intracellular antioxidant capability of HSF under oxidative stress brought on by H_2_O_2_ compared to the H_2_O_2_ injury model group. It is clear that the three dosages of 5.0 mg/mL, 10.0 mg/mL, and 20 mg/mL had a significant protective effect on cell survival (*p* < 0.001). Furthermore, compared to UF, LCF showed a more significant reduction in mortality due to H_2_O_2_ damage. Through cellular experiments, we chose these three concentrations for subsequent experiments.

### 3.5. Effects of UF and LCF on ROS Level

Figure 6 shows the intracellular ROS levels after H_2_O_2_ irradiation and incubation in both systems. Intracellular ROS levels rose after H_2_O_2_ injury, affecting the antioxidant capacity. There were significantly lower ROS levels after UF and LCF incubation in a dose-dependent manner compared to the H_2_O_2_ damage model. It is also clear that LCF has a much greater ability to reduce ROS levels than UF. At a concentration of 20.0 mg/mL, ROS content was 56,181 ± 2762, close to the level of the blank group. The experimental results show that both systems can effectively inhibit the increase in intracellular ROS, thereby slowing H_2_O_2_ damage.

### 3.6. Effects of UF and LCF on Cellular Antioxidant Capacity

The total antioxidant level, including different antioxidant compounds, enzymes, etc., is referred to as “total antioxidant capacity”. Figure 7 shows the total antioxidant capacity of HSF. The outcomes demonstrate that the total antioxidant capacity HSF was raised by both UF and LCF. However, after being incubated with LCF as opposed to UF, the total antioxidant capacity of HSF rose dramatically (*p* < 0.001) in a dose-dependent manner, peaking at a dosage of 20.0 ± 0.006 mM.

### 3.7. Effects of UF and LCF on Antioxidant Enzyme Content and mRNA Expression

Further research was conducted on the enzymatic activity and RNA expression levels of individual antioxidant enzymes in cells (Figure 8). The levels of SOD, CAT, and GSH-Px and their mRNAs were considerably decreased after H_2_O_2_ induction. After treatment with *Euryale ferox* samples, the content of these three enzymes and their mRNA expression were elevated, and the effect was more significant with LCF.

### 3.8. Effects of UF and LCF on Nrf2/Keap1 Signaling Pathway

The expression of numerous anti-inflammatory and antioxidant genes is regulated by Nrf2, an important transcription factor in controlling oxidative stress which is crucial for the body’s induction of the antioxidant response. Figure 9a shows that after H_2_O_2_ induction, the amount of Nrf2 protein in the nucleus was lowered by 50.24%, while Nrf2 translocation to the nucleus was prevented and it kept building up in the cytoplasm. At the same time, the RNA expression of Nrf2 was significantly reduced by nearly five times (Figure 9b). The Keap1 expression and protein level were both two times higher in the cells, which made Nrf2 signaling more challenging (Figure 9c,d). The nuclear translocation of Nrf2 and Keap1 was greatly enhanced following sample treatment in a dose-dependent manner. After fermentation, the impact of the samples on Nrf2 and Keap1 was considerably more notable. When the quantity of LCF was 10 mg/mL, the protein content in the nuclei of Keap1 and Nrf2 cells was comparable to that of the control group. The production of numerous downstream target proteins can start when the Keap1-Nrf2/ARE signaling pathway is activated. When activated, these target proteins can control the body’s redox balance and return oxidative stress to a normal physiological state. NQO-1 and HO-1 are the most important antioxidant proteins downstream of Nrf2, and their enzymatic activity increased differently in sample group HSF cells compared to the H_2_O_2_ and control groups, with their relative expression levels showing extremely significant differences (*p* < 0.001). The most significant effect was seen in NQO-1, the enzyme activity and gene expression of which generally increased 2–3-fold, with the highest level even reaching 5.61-fold. At the same time, LCF showed a greater capacity for improvement than UF. At a dosage of 10 mg/mL, in particular, both enzyme activity and expression levels displayed their greatest variation (*p* < 0.001). The GCLC protein content and gene expression levels were significantly increased after sample treatment (*p* < 0.001). A huge difference was also shown between UF and LCF (*p* < 0.001).

### 3.9. Effects of UF and LCF on MAPK Signaling Pathway

As shown in Figure 10, the extracellular regulated protein kinases (ERK) signaling pathway regulates cell growth and differentiation, while the c-Jun N-terminal kinase (JNK) and p38MAPK signaling pathways play important roles in inflammation and apoptosis, among other stress responses. We found that the range of the three proteins increased nearly two-fold after H_2_O_2_ induction. With increasing sample concentration, the enzyme content of the treated cells gradually decreased and approached that of the blank control group. In particular, the most significant effect was seen in p18. The most significant levels of the three proteins in LCF-treated cells were found at a concentration of 10 mg/mL (*p* < 0.01).

### 3.10. Effects of UF and LCF on Bax and Bcl-2 Protein Content

We assessed the effects of the samples on Bax and Bcl-2 in cells exposed to H_2_O_2_. According to the experimental findings (Figure 11a–d), the injured cells had considerably higher levels of Bax protein and mRNA expression. Bcl, on the other hand, had lower levels of mRNA expression and protein concentration.

After sample treatment, the two genes showed opposite effects. The three different sample concentrations significantly decreased the content and expression of Bax when compared to the oxidative stress model group (*p* < 0.001), while increasing the protein content and mRNA expression of Bcl, and the effect on both genes increased with the increase in sample concentration. The graphs (Figure 11e,f) unmistakably demonstrate the proportionate link between protein content and Bcl-2 and Bax mRNA expression levels. After sample treatment, there was a very high ratio of Bcl-2 to Bax in terms of both protein concentration and related mRNA expression. When the LCF concentration reached 10 mg/mL, the effect was significantly stronger than that of the blank control group. In addition, comparing the effect of UF with that of LCF, we found that the effect of LCF on both genes was much higher than that of UF, especially for Bcl, and the difference reached 2.4-fold.

### 3.11. Effects of UF and LCF on Content and Expression Level of MMP-1 and COL-1

The content and expression levels of MMP-1 and COL-1 are crucial physicochemical markers of anti-aging. Figure 12 displays the anti-aging effects of UF and LCF. After H_2_O_2_-induced damage, the levels of MMP-1 and COL-1 in HSF increased and decreased nearly threefold, respectively. Different sample concentrations led to a substantial increase in COL-1 content and RNA expression level as compared to the H_2_O_2_ group (*p* < 0.001), and the increase was dose-dependent. At concentrations of up to 2 mg/mL, COL-1 levels were 2.295 times (UF group) and 2.911 times (LCF) higher than those of the H_2_O_2_ group, respectively, approaching the results of the control group. In addition, compared to those of untreated cells, MMP-1 expression and enzymatic activity were significantly decreased (*p* < 0.001) by up to 33.2% (UF) and 40.6% (LCF), respectively. These results suggest that H_2_O_2_ induction increases intracellular MMPs, which in turn decreases COL-1 levels, leading to cellular damage and skin aging, specifically by degrading the extracellular matrix (ECM) components and breaking the normal structure of collagen and elastic fibers. However, UF and LCF were shown to significantly inhibit the overexpression of MMPs induced by H_2_O_2_, and effectively slow down and fight off accelerated aging in H_2_O_2_-induced HSF. Moreover, LCF was shown to have a stronger anti-aging effect than UF.

## 4. Discussion

Antioxidant research has received increasing attention in recent years, and several Chinese medicinal plants are thought to have high antioxidant potential. *Euryale ferox* is a plant of great interest by virtue of its pharmacological effects such as antioxidation and anti-aging properties and the improvement of hypoglycemia and myocardial ischemia. To investigate the functions and mechanisms of *Euryale ferox* in antioxidative stress and anti-aging, we constructed an HSF model of H_2_O_2_ damage. The results showed that treatment with *Euryale ferox* fermented by *L. curvatus* was more effective than treatment with unfermented *Euryale ferox* in reducing H_2_O_2_-induced oxidative stress and protecting HSF from H_2_O_2_-induced cytotoxicity. It was also discovered that *Euryale ferox* fermented by *L. curvatus* improved the Bcl-2/Bax ratio and decreased the amount of MMPs, which reduced apoptosis and skin senescence brought on by oxidative stress.

Among the bioactive components of *Euryale ferox*, the polyphenols and flavonoids were significantly increased after fermentation. Our results are consistent with the results of previous studies. After fermenting kiwifruit with Lactobacillus plantarum, Zhou et al. measured the levels of total phenols and total flavonoids and found that they were both higher [25]. Fermented *Euryale ferox* also had higher polysaccharide and polypeptide content than unfermented *Euryale ferox*, while also having a lower molecular weight. This is probably because, during the fermentation process, proteases secreted by lactic acid bacteria break down the large molecular weight proteins or polypeptides. Furthermore, it was discovered that fermented *Euryale ferox* polysaccharides showed structural variations. This is probably because *L. curvatus* fermentation breaks down *Euryale ferox* polysaccharides and polypeptides into lower molecular weight polysaccharides and polypeptides. In our earlier investigation, we confirmed that the molecular weight of Dendrobium officinale polysaccharides was reduced both before and after fermentation [26]. Changes in the content and compound level structure of bioactive substances during fermentation due to microbial metabolism could explain these findings. In addition, the structural integrity of the grain cell wall may be compromised by fermentation, which can lead to the release or synthesis of a variety of bioactive substances [8]. We also discovered that the in vitro antioxidant ability of *Euryale ferox* increased after fermentation, which may be connected to its altered bioactive components. Polyphenols are an important active ingredient in gorgonians, with a powerful ability to bind free radicals and reduce or even prevent oxidation processes altogether [27]. Studies have revealed that an increase in total phenolic compounds may be the cause of the observed increase in antioxidant activity capacity [28]. Berry polyphenol bioavailability has been observed to be low, but during fermentation, glucosidases have a positive influence on it, enhancing in situ ROS scavenging [29] and making it another phenolic substance in *Euryale ferox* flavonoids. According to previous studies, flavonoids interact with metals to form complexes that prevent metal-induced lipid oxidation and efficiently scavenge hydroxyl and peroxyl radicals [30]. Berries undergo biotransformation during fermentation, which speeds up the breakdown of phenolic chemicals and raises their bioavailability [29]. Additionally, the composition and size of polypeptides and polysaccharides have an impact on free radical scavenging. Studies have linked the size of polypeptide molecules to DPPH radical scavenging activity. Particularly at 3–5 kDa, low molecular weight (LMW) peptides have shown superior DPPH radical scavenging activity over high molecular weight (HMW) peptides [31]. This is consistent with the findings of previous studies [32,33]. According to previous studies, *Euryale ferox* polysaccharides have a scavenging effect on superoxide anions and hydroxyl radicals, and the intensity of the effect grows as the concentration of polysaccharides rises [34]. However, many naturally occurring polysaccharides are too large to penetrate cells and have an impact on biology, or some of their chemical structures have minimal activity [35]. In order to increase their biological activity, different microorganisms can be used to transform macromolecular compounds into small molecules [36,37]. Therefore, we can conclude that fermentation promoted changes in the content and structure of the bioactive components of *Euryale ferox*, leading to an increase in vitro antioxidant levels. The observed increase in total antioxidant activity may be attributed to the enhanced release of these antioxidant chemicals and the synergistic interactions between, for example, polyphenolic compounds and/or other extract components.

H_2_O_2_ is a ROS that can easily pass through cell membranes to cause the generation of free radicals and lipid peroxidation, which prevent cell growth and cause cellular senescence and death. Our results show an increase in the level of ROS in HSF cells after induction with H_2_O_2_. As antioxidant enzyme activity dramatically decreased and MMP activity increased, Bcl-2/Bax levels also significantly decreased. This can trigger a high oxidative stress response in the cells which harms the skin’s collagen and elastin fibers, speeds up the aging process, and even causes apoptosis. The body has enzymatic antioxidant systems such as GSH-Px, CAT, and SOD to maintain a dynamic redox balance of intracellular free radicals and ROS. These antioxidants can prevent oxidative damage to cells and tissues, neutralize free radicals and ROS, and limit the harm that they do to cells [38]. However, when oxidative stress brought on by ROS exceeds the ability of cellular antioxidant enzymes to scavenge free radicals, exogenous supplementation is required to reduce oxidative stress.

We first looked into the antioxidant enzyme activity in injured cells treated with *Euryale ferox* in order to understand the mechanism underlying the protective effects of LCF. According to the findings, fermented *Euryale ferox* lessens oxidative stress by reducing the formation of ROS and boosting the expression of oxidative stress genes, mitochondria, and antioxidants, including SOD, CAT, and GSH-Px. Superoxide anions are known to be broken down by SOD into the molecules O_2_ and H_2_O_2_. CAT then breaks down H_2_O_2_ into water and oxygen, contributing significantly to the preservation of intracellular redox equilibrium [1]. This is in line with recent research showing that fermenting blueberries and blackberries increases the production of antioxidant enzymes such as SOD, which enhance the removal of free radicals from the body [39]. We also discovered that cells exposed to *Euryale ferox* expressed more HO-1, NQO1 and GCLC mRNA than the control, and the effect was notably stronger with fermented *Euryale ferox*. This could be caused by HO-1 functions as an anti-inflammatory and antioxidant agent alongside CO and bilirubin to suppress apoptosis and stop free hemoglobin from taking part in oxidative reactions [40]. It is crucial to maintain the GSH supply to increase the activity of the enzymes involved in the GSH cycle (which is the basis for the cellular control of ROS) [41]. Studies have shown that ginseng extract protects human umbilical vein endothelial cells from H_2_O_2_-induced damage by enhancing the activity of GSH metabolizing enzymes, especially GSH-Px [42]. These data provide additional evidence that fermented *Euryale ferox* treatment enhances antioxidant enzyme activity, and promotes the synthesis of HO-1, GSH, and related enzymes, to reduce oxidative stress in cells.

ROS is a major cause of endogenous aging, and its accumulation can directly damage DNA, proteins, and lipids, triggering the downregulation of collagen production and leading to dermal aging. The AP-1 transcription factor complex upregulates MMPs and promotes collagen catabolism [43,44]. Thus, COL-I content and MMP-1 enzyme activity and expression levels are important physicochemical indicators of skin aging. After sample treatment, our experimental findings revealed a significant drop in MMP-1 enzyme content and expression levels along with an increase in COL-I. It has been found that many bioactive components of plant extracts, such as polyphenols, flavonoids, vitamins, carotenoids, and hydroxy acids have anti-aging properties [45]. Quercetin which is widely found in plants has excellent antioxidant activity and produces anti-aging effects on the skin through a variety of mechanisms: reduction in intracellular ROS, protection against oxidative damage to cells, downregulation of MMP-1 mRNA expression, and increased collagen production, as well as the upregulation of COL1A1 mRNA expression in HSF [46]. Rose hips flower extract protects collagen from degradation by inhibiting MMP-1 and MMP-9 expression and collagenase activity [47]. Our research has shown that fermented *Euryale ferox* maintains skin elasticity by down-regulating MMP-1 levels and up-regulating COL-I levels to prevent collagen aging and destruction.

The strength of apoptosis suppression has been discovered to be significantly influenced by the ratio of the Bax/Bcl-2 proteins. As a result, Bax is regarded as one of the most significant pro-apoptosis genes. By encouraging the expression of Bcl-2 and successfully limiting the expression of the pro-apoptotic protein Bax, fermented *Euryale ferox* raises the Bcl-2/Bax ratio and lowers apoptosis.

There are many endogenous antioxidant signaling pathways in the cell body, of which the Nrf2/Keap1/Mafk signaling-mediated expression of antioxidant enzymes enhances the cellular antioxidant system. Figure 13 illustrates how *Euryale ferox* regulates the HSF protection mechanism. In a healthy state, Nrf2 is primarily in the cytoplasm and linked to its inhibitor Keap1. ROS stimulation causes Nrf2 to decouple from Keap1, enter the nucleus, bind to the antioxidant response element, activate target genes, and control the transcriptional activity of oxidative enzymes, preventing damage caused by oxidative stress [48,49]. We can see that *Euryale ferox* and its fermentation promote the activation of Nrf2 signaling, which initiates the expression of a variety of downstream target proteins. NQO1 is a key phase II metabolic enzyme regulated by the Nrf2 signaling pathway. It catalyzes the reduction and destruction of hazardous quinones and their derivatives using NADH or DPH as electron donors, thereby preventing their continued involvement in redox processes and the formation of ROS [50]. We clearly observed a considerable increase in the content and expression of NQO1 protein under the control of Nrf2 signaling. Major antioxidant proteases including HO-1, SOD, CAT, and GSH-Px are also controlled by the Nrf2 signaling pathway. In the aforementioned trials, it was discovered that Nrf2 signaling greatly boosted the levels of SOD, CAT, and GSH-Px. HO-1 mRNA and protein expression are upregulated after oxidative stress and cellular damage, while Nrf2 can also directly regulate HO-1 promoter activity [40]. Polyphenols can upregulate the antioxidant response pathway mediated by transcription factor Nrf2, mitochondrial MnSOD, etc., thus promoting the expression of antioxidant enzymes [51,52].

In addition, we found that fermented *Euryale ferox* could slow down oxidative stress caused by ROS accumulation and prevent cellular stress responses such as inflammation and apoptosis by regulating the/P38/JUK/ERK signaling pathway. This is in line with studies on other natural substances (apigenol, ginsenosides, and apigenin) and polyphenols such as quercetin, resveratrol and tea polyphenols [51,52,53].

In addition, antioxidative stress and anti-inflammatory functions are usually formed by the interplay between different pathways and mechanisms and cannot be attributed to only one mechanism of action. Numerous studies have demonstrated that the Nrf2 and MAPK signaling pathways interact, and that Nrf2 activity and HO-1 synthesis can be increased by inhibiting MAPK signaling. In turn, Nrf2 activity can control how the MAPK signaling pathway is activated [54,55]. Baicalin, one of the main bioactive flavonoids found in the roots of the medicinal plant S. baicalensis, has been demonstrated to both activate and inhibit the MAPK signaling pathway and Nrf2 signaling pathway, thereby relieving upstream oxidative stress and inflammation [56]. Thus, the ability of fermented *Euryale ferox* to resist oxidative stress may be the result of the interaction of multiple signaling pathways, including Nrf2 and MAPK.

## 5. Conclusions

Our research shows that fermentation with *L. curvatus* significantly improves the bioactive components of *Euryale ferox*. Based on biochemical, cellular, and molecular studies, unfermented and fermented *Euryale ferox* is also shown to be extremely protective against H_2_O_2_-induced oxidative stress damage to HSF and attenuate apoptosis and senescence caused by oxidative stress in HSF. *L. curvatus* has also demonstrated success in fermentation with many other plant varieties, and it possesses great potential for future study.

## Figures and Tables

**Figure 1 antioxidants-11-01881-f001:**
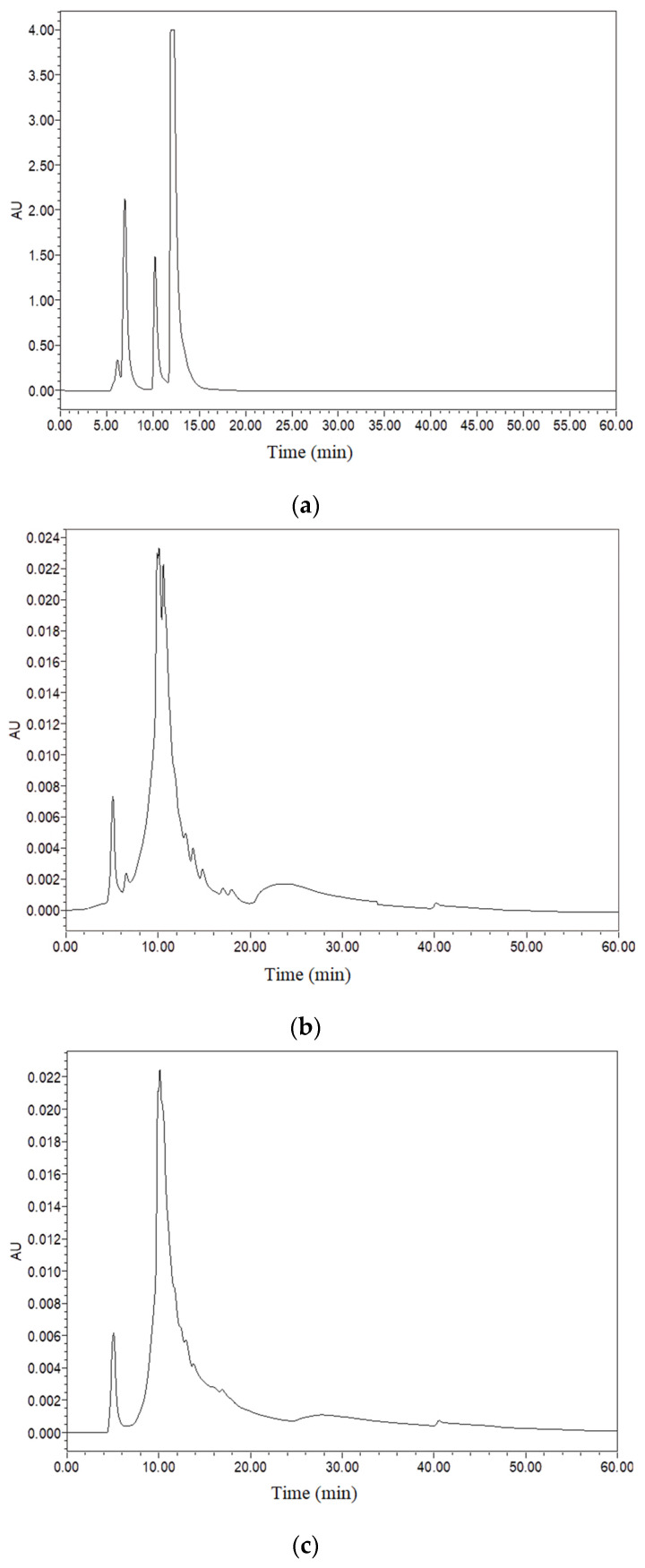
UF and LCF polypeptide molecular weight analysis using high−performance liquid chromatography (HPLC): (**a**) combined standards of bovine serum albumin, vitamin B12, and oxidized glutathione underwent HPLC analysis; (**b**) HPLC analysis of UF; (**c**) HPLC analysis of LCF. UF: unfermentation; LCF: single microbial fermentation with *L. curvatus*.

**Figure 2 antioxidants-11-01881-f002:**
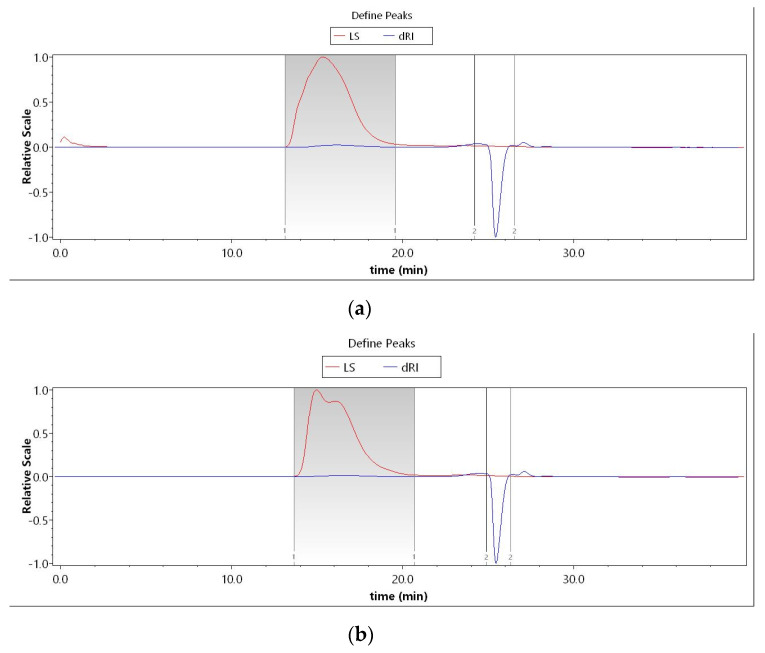
UF and LCF polysaccharide molecular weight analysis using GPC: (**a**) GPC analysis of UF; (**b**) GPC analysis of LCF. UF: unfermentation; LCF: single microbial fermentation with *L. curvatus*.

**Figure 3 antioxidants-11-01881-f003:**
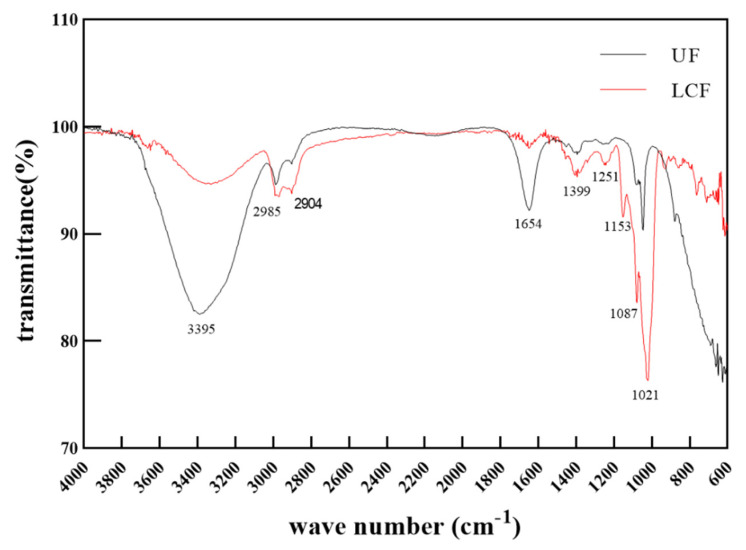
FT−IR spectra of UF and LCF. UF: unfermentation; LCF: single microbial fermentation with *L. curvatus*.

**Figure 4 antioxidants-11-01881-f004:**
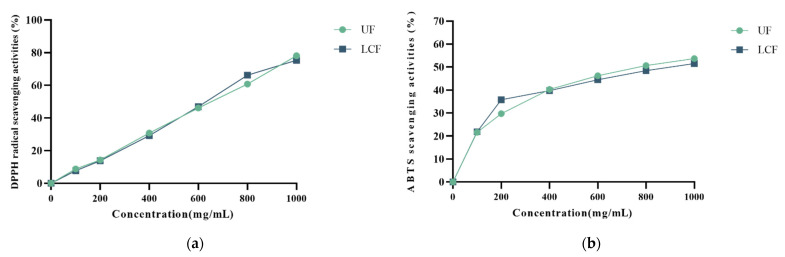
Phenolic components from samples of *Euryale ferox* show DPPH (**a**) and ABTS (**b**) scavenging properties. UF: unfermentation; LCF: single microbial fermentation with *L. curvatus*.

**Figure 5 antioxidants-11-01881-f005:**
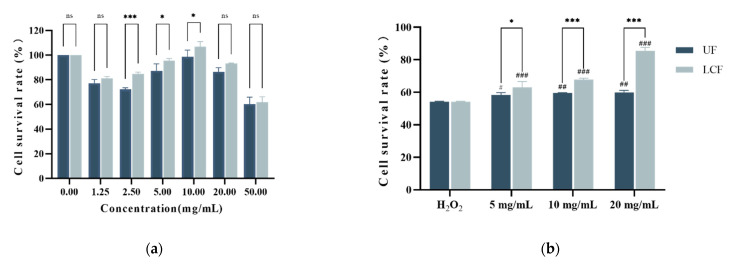
Effects of UF and LCF on HSF cell viability: (**a**) cytotoxicity of UF and LCF; (**b**) protective effects of UF and LCF on H_2_O_2_-damaged HSF; UF: unfermentation; LCF: single microbial fermentation with *L. curvatus*. Analysis of significant differences between each group and H_2_O_2_ model group, #: *p* < 0.05; ##: *p* < 0.01; ###: *p* < 0.001.; comparison of each group between UF and LCF, *: *p* < 0.05; ***: *p* < 0.001; ns: *p* > 0.05.

**Figure 6 antioxidants-11-01881-f006:**
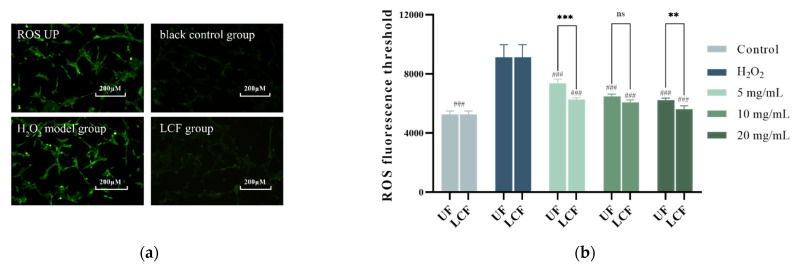
Effects of UF and LCF on ROS level: (**a**) H_2_O_2_-induced ROS production in cells; (**b**) intracellular DCFH-DC fluorescence threshold. UF: unfermentation; LCF: single microbial fermentation with *L. curvatus*. Analysis of significant differences between each group and H_2_O_2_ model group; ###: *p* < 0.001; ns: *p* > 0.05.comparison of each group between UF and LCF, **: *p* < 0.01; ***: *p* < 0.001; ns: *p* > 0.05.

**Figure 7 antioxidants-11-01881-f007:**
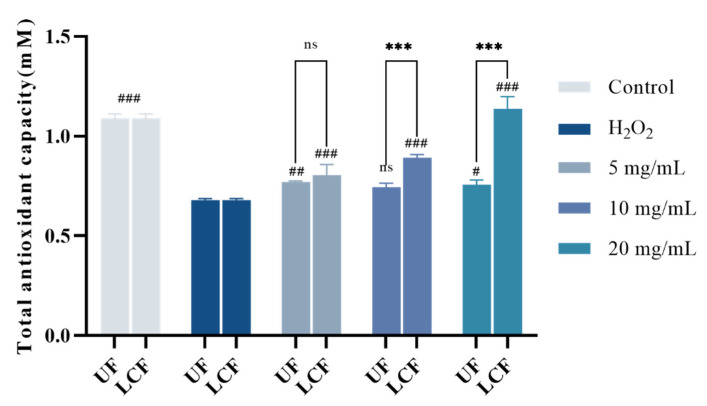
Effects of UF and LCF on cellular antioxidant capacity. UF: unfermentation; LCF: single microbial fermentation with *L. curvatus*. Analysis of significant differences between each group and H_2_O_2_ model group, #: *p* < 0.05; ##: *p* < 0.01; ###: *p* < 0.001. comparison of each group between UF and LCF, ***: *p* < 0.001; ns: *p* > 0.05.

**Figure 8 antioxidants-11-01881-f008:**
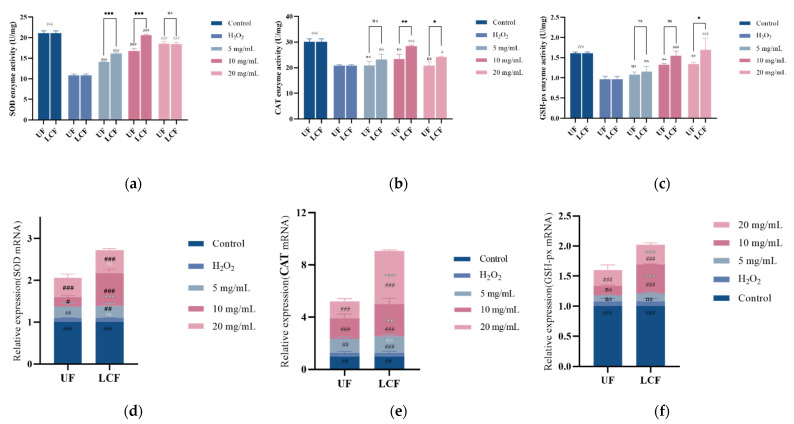
Effects of UF and LCF on antioxidant enzyme content and mRNA expression: (**a**,**d**) SOD content and mRNA expression; (**b**,**e**) CAT content and mRNA expression; (**c**,**f**) GSH-Px content and mRNA expression. UF: unfermentation; LCF: single microbial fermentation with *L. curvatus*. Analysis of significant differences between each group and H_2_O_2_ model group, #: *p* < 0.05; ##: *p* < 0.01; ###: *p* < 0.001.; comparison of each group between UF and LCF, *: *p* < 0.05; **: *p* < 0.01; ***: *p* < 0.001; ns: *p* > 0.05.

**Figure 9 antioxidants-11-01881-f009:**
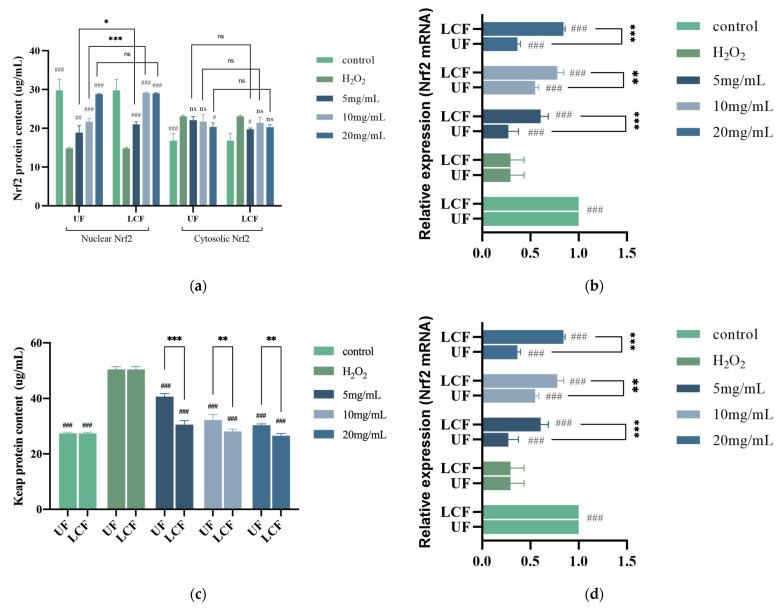
Effects of UF and LCF on Nrf2/Keap1 signaling pathway. Effects of UF and LCF on Nrf2 content (**a**) and mRNA expression (**b**), and Keap1 content (**c**) and mRNA expression (**d**). Effects of UF and LCF on activity of NQO-1 (**e**) and mRNA expression (**f**), HO-1 content (**g**) and mRNA expression (**h**), and GCLC content (**i**) and mRNA expression (**j**). UF: unfermentation; LCF: single microbial fermentation with *L. curvatus*. Analysis of significant differences between each group and H_2_O_2_ model group, #: *p* < 0.05; ##: *p* < 0.01; ###: *p* < 0.001.; comparison of each group between UF and LCF, *: *p* < 0.05; **: *p* < 0.01; ***: *p* < 0.001; ns: *p* > 0.05.

**Figure 10 antioxidants-11-01881-f010:**
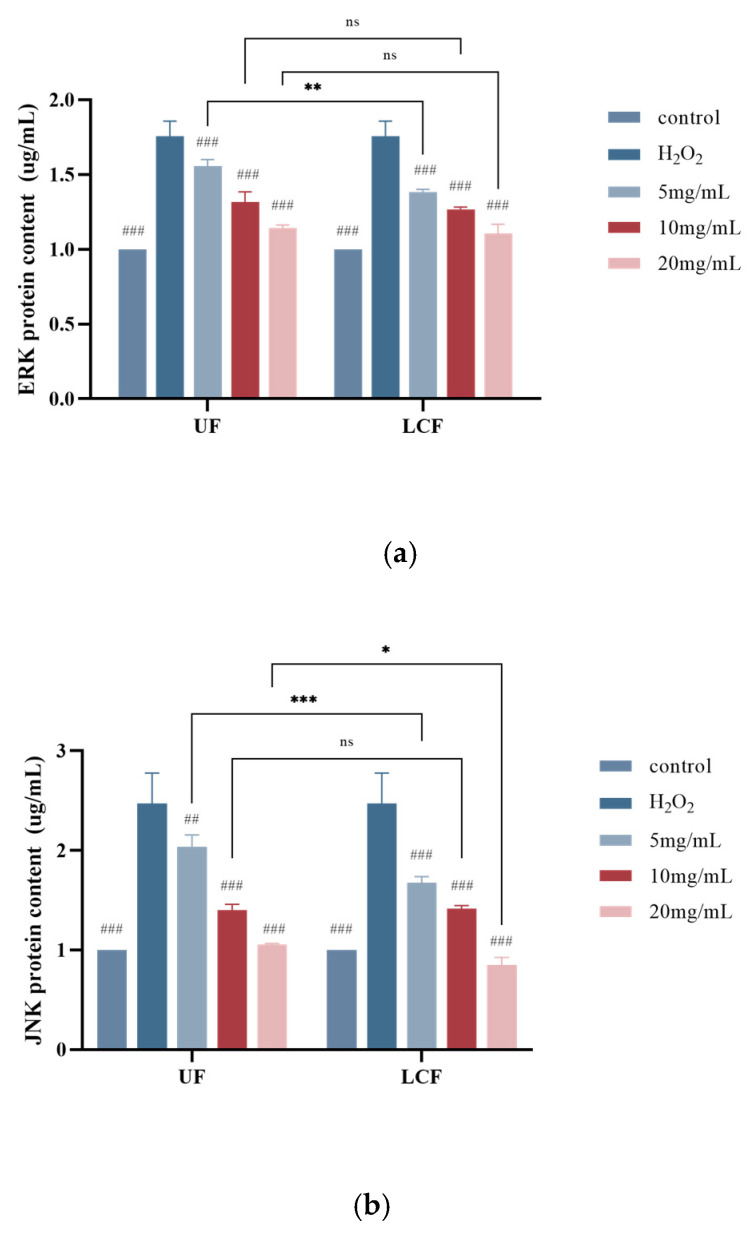
Effects of UF and LCF on MAPK signaling pathway: (**a**) effects on ERK protein content; (**b**) effects on JNK protein content; (**c**) effects on p38 protein content. UF: unfermentation; LCF: single microbial fermentation with *L. curvatus*. Analysis of significant differences between each group and H_2_O_2_ model group, ##: *p* < 0.01; ###: *p* < 0.001.; comparison of each group between UF and LCF, *: *p* < 0.05; **: *p* < 0.01; ***: *p* < 0.001; ns: *p* > 0.05.

**Figure 11 antioxidants-11-01881-f011:**
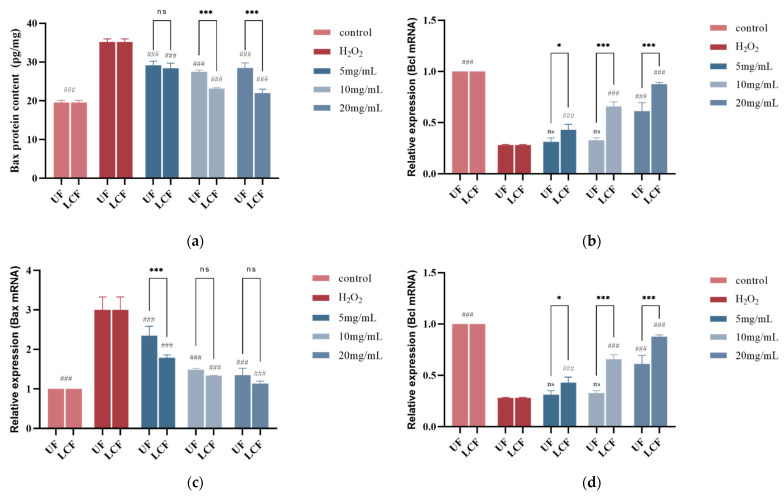
(**a**,**c**) Effects of UF and LCF on content and expression level of Bax and mRNA; (**b**,**d**) effects on content and expression level of Bcl-2 and mRNA; (**e**,**f**) effects on Bax/Bcl-2 ratio. UF: unfermentation; LCF: single microbial fermentation with *L. curvatus*. Analysis of significant differences between each group and H_2_O_2_ model group, ###: *p* < 0.001.; comparison of each group between UF and LCF, *: *p* < 0.05; ***: *p* < 0.001; ns: *p* > 0.05.

**Figure 12 antioxidants-11-01881-f012:**
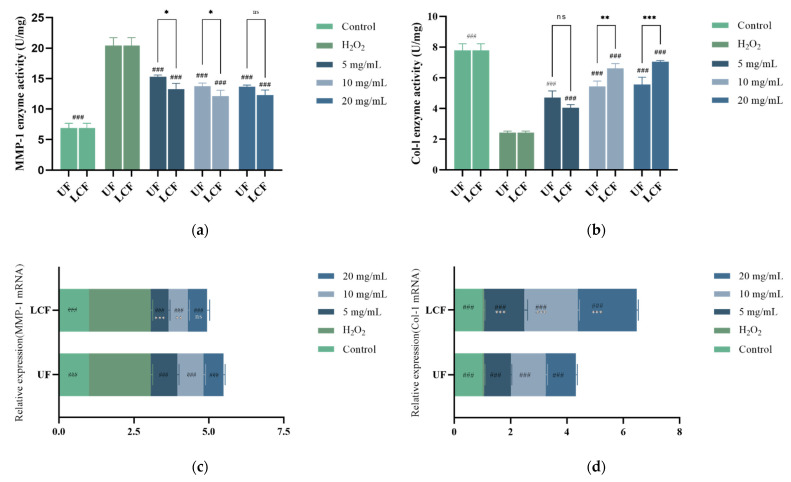
Effects of UF and LCF on content and expression level of MMP-1 and COL-1: (**a**,**c**) effects on content and expression level of MMP-1 and mRNA; (**b**,**d**) effects on content and expression level of COL-1 and mRNA. UF: unfermentation; LCF: single microbial fermentation with *L. curvatus*. Analysis of significant differences between each group and H_2_O_2_ model group, ###: *p* < 0.001.; comparison of each group between UF and LCF, *: *p* < 0.05; **: *p* < 0.01; ***: *p* < 0.001; ns: *p* > 0.05.

**Figure 13 antioxidants-11-01881-f013:**
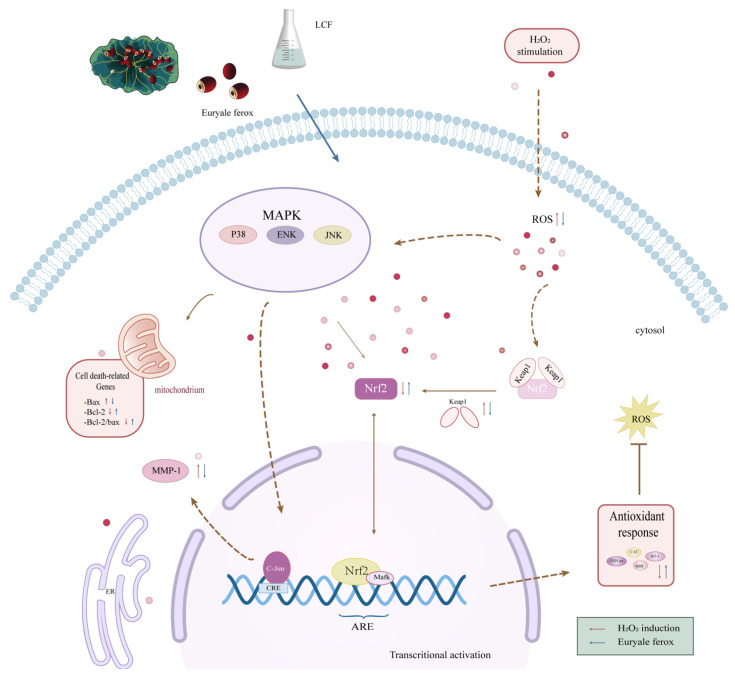
The oxidative stress pathway of Nrf2/Keap1/Mafk.

**Table 1 antioxidants-11-01881-t001:** Major *Euryale ferox* bioactive component variations across various fermentation processes.

Bioactive Components	Fermentation System (mg/mL)	
	UF	LCF
Total phenolic content	0.041 ± 0.000 ^a^	0.052 ± 0.015 ^b^
Total protein content	4.320 ± 0.700 ^a^	7.463 ± 0.261 ^a^
Total flavonoid content	1.358 ± 0.167 ^a^	1.402 ± 0.000 ^b^
Total polypeptide content	0.399 ± 0.009 ^a^	0.627 ± 0.012 ^a^
Total polysaccharide content	1.657 ± 0.088 ^a^	2.101 ± 0.000 ^b^

UF: unfermentation; LCF: single microbial fermentation with *L. curvatus*. Means in the same row with different lowercase characters (^a^,^b^) differ considerably (*p* = 0.05).

**Table 2 antioxidants-11-01881-t002:** Distribution of molecular weights of polypeptides in UF and LCF.

Sample Name	Peak Limits (min)	Area (%)	Molecular Weight (Da)
UF	5.10	7.21	350,018.87
	10.13	92.79	2539.43
LCF	5.10	10.24	349,156.22
	9.92	30.37	3123.51
	10.10 10.59	23.17 36.23	2608.02 1625.89

UF: unfermentation; LCF: single microbial fermentation with *L. curvatus*.

**Table 3 antioxidants-11-01881-t003:** Distribution of molecular weights of polysaccharides in UF and LCF.

	UF		LCF	
Peak Name	Peak 1	Peak 2	Peak 1	Peak 2
**Peak Limits (min)**	13.131–19.579	24.210–26.526	13.629–20.683	24.887–26.312
**Mw**	1.617 × 10^6^ (±1.946%)	5.099 × 10^4^ (±20.538%)	1.088 × 10^6^ (±1.088%)	4.306 × 10^3^ (±13.482%)
**Mz**	1.139 × 10^7^ (±8.631%)	1.612 × 10^5^ (±47.918%)	4.140 × 10^6^ (±2.591%)	3.382 × 10^4^ (±40.654%)
**Mw/Mn**	1.809 (±2.285%)	2.259 (±26.791%)	1.631 (±1.391%)	1.851 (±17.937%)
**Mz/Mn**	12.741 (±8.714%)	7.143 (±50.912%)	6.204 (±2.732%)	14.536 (±42.340%)

UF: unfermentation; LCF: single microbial fermentation with *L. curvatus*.

## Data Availability

Data is contained within the article.

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
