# Peer review of "Enhancing Bioactive Components of Euryale ferox with Lactobacillus curvatus to Reduce H2O2-Induced Oxidative Stress in Human Skin Fibroblasts"

_antioxidants, 2022, doi:10.3390/antiox11101881_

Round 1
Reviewer 1 Report
The manuscript by Jiang describes the impact of fermentation on E.ferox ameliorating activity in HDF. The authors use several common methodologies to monitor H2O2-induced oxidative damage and mRNA observation.
The manuscript lacks several crucial points
1. The rational of plant selection is not clear and its activity is at a high nonrelevant concentration of mg/ml.
2. The impact of fermentation on bioavailability is not new. The author should demonstrate the effect in three independent plants type and to see if the enhance anti-oxidant activity is due to the method or specifically to this plant
3. A crucial missing control – the authors compared the effect of unfermented and fermented plants. The bacteria itself can add background to the majority of assays, including the analytical methods. In addition, compounds secreted from bacteria had been shown to activate NRF2 signaling and modulate the cellular defense (e.g. Saturated and aromatic aldehydes originating from skin and cutaneous bacteria activate the Nrf2-keap1 pathway in human keratinocytes - PubMed (nih.gov)). The minor impact in the bioassay may b due to bacterial secretion.
4. At least 1-3 specific antioxidant molecules enhanced by the fermentation should be presented and quantified (what polyphenol)
Author Response
Detailed response to reviewer 1’ comments
September 4, 2022
Journal: Antioxidants
Manuscript Number: antioxidants-1883338
Title: “Enhancing Bioactive Components of Euryale ferox with Lactobacillus curvatus to Reduce H2O2-induced Oxidative Stress in Human Skin Fibroblasts”
Authors:, YanBing Jiang , Shiquan You , Yongtao Zhang , Jingsha Zhao , Dongdong Wang , Dan Zhao , Meng Li * ChangTao Wang
Dear Editor and reviewer,
Thank you very much for giving us this opportunity to revise our manuscript. Your comments and suggestions on our manuscript have been encouraging and helpful. The following pages provide a detailed point-by-point response to your comments. Note that the reviewer’ comments are presented in Italics, and our responses are in Roman and blue font. In addition, we addressed all these major points and other issues carefully and revised the manuscript accordingly, and We highlight the revised parts in the article in yellow. Please let me know if you have any further questions.
In response to Reviewer 1's suggestion, we have made the following modifications and responded:
- The rational of plant selection is not clear and its activity is at a high nonrelevant concentration of mg/ml?
Reply:Thanks to the reviewer for reviewing this paper and for the question, We chose Euryale ferox because the plant is rich in polyphenols, flavonoids and other ingredients with antioxidant properties, which have been shown to have antioxidant and free radical scavenging effects. I will add more details in the introduction section.
- The impact of fermentation on bioavailability is not new. The author should demonstrate the effect in three independent plants type and to see if the enhance anti-oxidant activity is due to the method or specifically to this plant.
Reply:Thanks to the reviewer's question. Our experiment was carried out independently on this plant, with different strains of fermentation. The Lactobacillus curvatus used in this paper was selected from a comparative experiment of three strains, the other two strains are not presented in this paper.
- A crucial missing control – the authors compared the effect of unfermented and fermented plants. The bacteria itself can add background to the majority of assays, including the analytical methods. In addition, compounds secreted from bacteria had been shown to activate NRF2 signaling and modulate the cellular defense (e.g. Saturated and aromatic aldehydes originating from skin and cutaneous bacteria activate the Nrf2-keap1 pathway in human keratinocytes - PubMed (nih.gov)). The minor impact in the bioassay may b due to bacterial secretion.
Reply:Thanks to the reviewer's suggestion. Increased antioxidative activity in fermented plant-based foods is due to an increase in the release of antioxidant compounds. The improvement of antioxidative activity by fermentation is mainly due to an increase in phenolic compounds and flavonoids via microbial hydrolysis. Bacteria have a certain increasing effect but not as strong a capacity . [1] S. J. Hur, S. Y. Lee, Y. C. Kim, I. Choi, and G. B. Kim, “Effect of fermentation on the antioxidant activity in plant-based foods,” Food Chem., vol. 160, pp. 346–356, Oct. 2014, doi: 10.1016/J.FOODCHEM.2014.03.112.
In addition, this bacterium, which we have divided ourselves, is more capable of degrading mainly against polysaccharides (unpublished)
- At least 1-3 specific antioxidant molecules enhanced by the fermentation should be presented and quantified (what polyphenol)
.
Reply: Thanks to the reviewer's suggestion . In our experiments we measured the changes in the content of antioxidant active molecules such as polyphenols, total flavonoids and polysaccharides in Euryale ferox before and after fermentation. The total content of these molecules can represent the antioxidant capacity of the plant.
Sincerely,
Meng Li
Beijing Key Lab of Plant Resource Research and Development, Beijing Technology and Business University, Fucheng Road, Beijing 100048, China
Tel.: +86-13426015179
- Email: limeng@btbu.edu.cn
Reviewer 2 Report
This paper described the enhanced antioxidant of Euryale ferox after fermented with Lactobacillus curvatus. The experiment was well organized and the data is solid. but I think it need some improvement. I listed my suggestions as below.
1. Abstract: please use exact verb tense. A past tense should be used for what you did.
2. The genus and species name (e.g. Lactobacillus curvatus) should be italic
3. Section 2.3: After fermentation, did you check the growth or survival of Lactobacillus curvatus in the culture broth? How do you sure the water extract was fermented? When you prepare the fermented water extract, do you remove the bacteria first?
4. Line 119, The solution to be analyzed was first diluted to 3 mg/mL and… I can’t understand here. Do you mean you dried the water extract and redissolved in water at 3mg/ml?
5. Section 2-10, This section is hard to follow, please modify. Besides, please note the unit. I think it should be uL, not mL. Line 153, What does A, B and C present?
6. Figure 1, 2, the letter is too small to distinguish.
7. Figure 3, the value for y-axis is missing.
8. Line 106, 60-70º should be 60-70ºC
9. Line 139, 8x104 -1x105
10. Figure 4, Line 253, the author said “LCF was more effective than UF……” This is not true, the two line seem similar, and there is not marked with significant difference.
11. Figure 6, the letter in the image cannot be distinguished. All the figure and picture need to be modified.
12. Figure 8, compared with control, H2O2 group seems to increased.
Author Response
Detailed response to reviewer 2’ comments
September 4, 2022
Journal: Antioxidants
Manuscript Number: antioxidants-1883338
Title: “Enhancing Bioactive Components of Euryale ferox with Lactobacillus curvatus to Reduce H2O2-induced Oxidative Stress in Human Skin Fibroblasts”
Authors: YanBing Jiang , Shiquan You , Yongtao Zhang , Jingsha Zhao , Dongdong Wang , Dan Zhao , Meng Li *,ChangTao Wang
Dear Editor and reviewer,
Thank you very much for giving us this opportunity to revise our manuscript. Your comments and suggestions on our manuscript have been encouraging and helpful. The following pages provide a detailed point-by-point response to your comments. Note that the reviewer’ comments are presented in Italics, and our responses are in Roman and blue font. In addition, we addressed all these major points and other issues carefully and revised the manuscript accordingly, and We highlight the revised parts in the article in yellow. Please let me know if you have any further questions.
In response to Reviewer 1's suggestion, we have made the following modifications and responded:
- Abstract: please use exact verb tense. A past tense should be used for what you did
Reply:Thanks to the reviewer for reviewing this paper and for the question.We
have read the summary section carefully and there was indeed a tense error. We have reworked to correct it.
2.The genus and species name (e.g. Lactobacillus curvatus) should be italic. Line 106, 60-70º should be 60-70ºC. Line 139, 8x104 -1x105.
Reply:Thanks to the reviewer's question. These were our mistakes and we have corrected the wording.
3.Section 2.3: After fermentation, did you check the growth or survival of Lactobacillus curvatus in the culture broth? How do you sure the water extract was fermented? When you prepare the fermented water extract, do you remove the bacteria first? Line 119, The solution to be analyzed was first diluted to 3 mg/mL and… I can’t understand here. Do you mean you dried the water extract and redissolved in water at 3mg/ml?
Reply:Thanks to the reviewer's suggestion, we made an error in the description of "2.3. Fermentation with Lactobacillus curvatus" in the methods section. We have corrected it in the manuscript. We are freeze-drying the supernatant obtained after centrifugation of the fermentation extract and subsequently performing the subsequent experiments.
Section 2-10, This section is hard to follow, please modify. Besides, please note the unit. I think it should be uL, not mL. Line 153, What does A, B and C present?
Reply: We thank the reviewers for their suggestions. We have carefully considered your suggestions and have decided to revise the content of section 2.3.
The letters in the formula represent the following. A: Absorbance of sample wells; B: Absorbance of blank (no sample added); C: Absorb-ance of blank control (no cells added)
- Figure 1, 2, the letter is too small to distinguish. Figure 3, the value for y-axis is missing.Figure 6, the letter in the image cannot be distinguished. All the figure and picture need to be modified.
Reply: Thanks to the reviewer's suggestion. We have carefully considered your suggestions and have decided to modify all graphics and images.
- Figure 8, compared with control, H2O2 group seems to increased.
Reply:Thanks to the reviewer's suggestion. We have read your suggestions carefully. The levels of SOD, CAT and GSH-Px and their mRNAs were significantly lower in the H2O2 group in Figure 8.
- Figure 4, Line 253, the author said “LCF was more effective than UF……” This is not true, the two line seem similar, and there is not marked with significant difference
Reply: Thanks to the reviewer's suggestion. We thank the reviewers for their suggestions, and we have carefully considered your suggestions and decided to revise the conclusion.
Sincerely,
Meng Li
Beijing Key Lab of Plant Resource Research and Development, Beijing Technology and Business University, Fucheng Road, Beijing 100048, China
Tel.: +86-13426015179
- Email: limeng@btbu.edu.cn

Reviewer 3 Report
This paper presented enhancing bio-activities of Euryale ferox fermented with Lactobacillus strain to reduce H2O2-induced oxidative stress in human skin fibroblast. This article may be helpful for development of new fermentation of plant extracts as basic information. However, some revisions are necessary for better quality of this article including mimic errors as follows.
1. Line 48-50, this part need some references.
2. Line 101, the absorbance of broth
3. You used Euryale ferox water extract for fermentation. Please explain the principles on the increment of polyphenol contents in extraction by fermentation.
4. The m.w. of peptides was reduced for fermentation in your study. Does Lactobacillus strain has protease activity?
5. I think the `Discussion' part is too long. It looks like a review article. Please shorten this part.
6. Some mimic errors were detected. Please check your manuscript more carefully.
Ex) Line 106, insert `°C’. Line234, CONH2 → CONH2, all cm-1 → cm-1 etc.
Author Response
Detailed response to reviewer 3’ comments
September 4, 2022
Journal: Antioxidants
Manuscript Number: antioxidants-1883338
Title: “Enhancing Bioactive Components of Euryale ferox with Lactobacillus curvatus to Reduce H2O2-induced Oxidative Stress in Human Skin Fibroblasts”
Authors: YanBing Jiang , Shiquan You , Yongtao Zhang , Jingsha Zhao , Dongdong Wang , Dan Zhao , Meng Li * , ChangTao Wang
Dear Editor and reviewer,
Thank you very much for giving us this opportunity to revise our manuscript. Your comments and suggestions on our manuscript have been encouraging and helpful. The following pages provide a detailed point-by-point response to your comments. Note that the reviewer’ comments are presented in Italics, and our responses are in Roman and blue font. In addition, we addressed all these major points and other issues carefully and revised the manuscript accordingly, and We highlight the revised parts in the article in yellow. Please let me know if you have any further questions.
In response to Reviewer 1's suggestion, we have made the following modifications and responded:
- Line 48-50, this part need some references.
Reply: Thank you for the reviewer's suggestion. We have carefully considered your suggestion and have made the appropriate additions to this section..
- Line 101, the absorbance of broth
Reply: Thanks to the reviewer's suggestion, the use of "broth" in the description was an error on our part and we have corrected the wording to "Bacterial suspensions".
- You used Euryale ferox water extract for fermentation. Please explain the principles on the increment of polyphenol contents in extraction by fermentation.
Reply: Thanks to the reviewer's suggestion, we made an error in the description of "2.3. Fermentation with Lactobacillus curvatus" in the methods section. We ferment directly from the Euryale ferox, not from the Euryale ferox water extract.
The increase in polyphenol content in fermentation extraction may be due to: (1) extraction by fermentation increases the extraction rate compared to extraction with direct water; (2) microbial enzymes produced by fermentation, such as glucosidase, amylase, cellulase, chitinase, inulinase, phytase, xylanase, tannase, esterase, convertase or lipase, hydrolyse glucosides and break down plant cell walls or starch. These enzymes play a role in breaking down the plant cell wall matrix, thus facilitating the extraction of flavonoids. The presence of lactic acid bacteria in controlled fermentations facilitates simple phenolic conversion and the depolymerisation of high molecular weight phenolic compounds.
[1] S. J. Hur, S. Y. Lee, Y. C. Kim, I. Choi, and G. B. Kim, “Effect of fermentation on the antioxidant activity in plant-based foods,” Food Chem., vol. 160, pp. 346–356, Oct. 2014, doi: 10.1016/J.FOODCHEM.2014.03.112.).
- The m.w. of peptides was reduced for fermentation in your study. Does Lactobacillus strain has protease activity?
Reply: Thank you for the reviewer's suggestion. During fermentation, proteases secreted by lactic acid bacteria break down proteins or peptides of large molecular weight..
- I think the `Discussion' part is too long. It looks like a review article. Please shorten this part.
Reply: Thank you for the reviewer's suggestion.. We have carefully considered your suggestions and have decided to trim the discussion section.
- Some mimic errors were detected. Please check your manuscript more carefully.
Ex) Line 106, insert `°C’. Line234, CONH2 → CONH2, all cm-1 → cm-1 etc
Reply: Thank you for the reviewer's suggestion. A number of parody errors in the description were our mistake and we have corrected them in the text.
Sincerely,
Meng Li
Beijing Key Lab of Plant Resource Research and Development, Beijing Technology and Business University, Fucheng Road, Beijing 100048, China
Tel.: +86-13426015179
- Email: limeng@btbu.edu.cn

Round 2
Reviewer 1 Report
The main comments and missing control had not been adressed.
Author Response
September 14, 2022
Journal: Antioxidants
Manuscript Number: fantioxidants-1883338
Title: “Enhancing Bioactive Components of Euryale ferox with Lactobacillus curvatus to Reduce H2O2-induced Oxidative Stress in Human Skin Fibroblasts”
Authors: Meng Li * , YanBing Jiang , Shiquan You , Yongtao Zhang , Jingsha Zhao , Dongdong Wang , Dan Zhao , ChangTao Wang
Dear Editor and reviewer,
Thank you very much for giving us this opportunity to revise our manuscript. Your comments and suggestions on our manuscript have been encouraging and helpful. Below is a detailed point-by-point response to your comments. If you have any further questions, please let me know.
For the effect of the metabolic components of the strain itself, it is not possible to set up a perfect blank control for the strain with previous experimental experience. If the MRS medium is used as the blank group for the strain, it is also impossible to account for this, because the strain also uses the components of MRS as nutrients to catabolize, and most of the material obtained at the end is also the material after MRS has been catabolized, so the 'blank indicator' measured at the end is not really a blank in the sense. By the same token, our experiments were designed mainly with Euryale ferox as the main body, using the bacteria to catabolise and metabolise the gorgonian components, and the substances obtained in the end were mainly the fermented substances of Euryale ferox.
Sincerely,
Meng Li
Beijing Key Lab of Plant Resource Research and Development, Beijing Technology and Business University, Fucheng Road, Beijing 100048, China
Tel.: +86-13426015179
- Email: limeng@btbu.edu.cn
